Identifying and characterizing Stagonosporopsis cucurbitacearum causing spot blight on Pinellia ternata in China

Zhou Jia
Xu Jiawei
Xu Rong
Chen Qiaohuan
Wang Yunhan
Huang Bisheng
Liu Dahui
Miao Yuhuan miaoyh@hbtcm.edu.cn
Pharmacy faculty, Hubei University of Chinese Medicine , Wuhan , China
Franco Bernardo
Electronic publication date: 2022 Apr 13
Publication date: 2022
Volume: 10
Electronic Location ID: e13278
Received 2022 Jan 13; Accepted 2022 Mar 24
Copyright: © 2022 Zhou et al.
Copyright year: 2022
Copyright holder: Zhou et al.
License: This is an open access article distributed under the terms of the Creative Commons Attribution License, which permits unrestricted use, distribution, reproduction and adaptation in any medium and for any purpose provided that it is properly attributed. For attribution, the original author(s), title, publication source (PeerJ) and either DOI or URL of the article must be cited.
License URL: https://creativecommons.org/licenses/by/4.0/

Keywords: Pinellia ternata, Fungal pathogen, Spot blight, Stagonosporopsis cucurbitacearum

Funding: Major Increase and Decrease Projects at the Central Level of China 2060302 Science and Technology Innovation Special Project of Hubei Province 2019ACA119 Funding was supported by the Major Increase and Decrease Projects at the Central Level of China (2060302) and Science and Technology Innovation Special Project of Hubei Province (2019ACA119). The funders had no role in study design, data collection and analysis, decision to publish, or preparation of the manuscript.

==============================
Background

Pinellia ternata (Thunb.), a perennial herbal plant in the Araceae family, has great medicinal value and market demand. In August 2020, an outbreak of severe leaf spot blight disease resulted in a huge yield loss of P. ternata. It is necessary to isolate and identify the pathogens that cause spot blight on P. ternata.

Methods

In this study, we isolated and identified the pathogens by fulfilling Koch’s postulates. Disease samples with typical spot blight symptoms were collected and pathogens were isolated from the diseased tissues. The pathogen was identified based on its biological characteristics and molecular analysis of internal transcribed (rDNA-ITS) and large subunit (LSU) sequences. Phylogenetic tree were constructed using MEGA7 software and pathogenicity tests were performed using in vivo inoculation. Finally, the pathogen was recovered and identified from the inoculated plants.

Results

Based on Koch’s postulates, we identified the pathogen causing spot blight on P. ternata as Stagonosporopsis cucurbitacearum. To our knowledge, this is the first study to explore spot blight on P. ternata caused by S. cucurbitacearum in China.

Introduction

Pinellia ternata (Thunb.) is a perennial herbal plant in the Araceae family that has expectorant, antitussive and antiemetic functions (Zuo & Mou, 2019). The amazing medicinal valves of P. ternata are attributed to the large number of secondary metabolites, including alkaloids, organic acids, volatile oil, and flavonoids, in its tubers (Ding, Song & Hu, 2021). Because of its high medicinal value, the P. ternata tuber is widely planted and used in many provinces of China, including Hubei, Henan, Guizhou, and Gansu. P. ternata have been used clinically in traditional Chinese medicine (TCM) for centuries. It is one of the 21 traditional Chinese Lung Cleansing and Detoxifying Decoction medicines used to treat the symptoms of COVID-19, and the key role of P. ternata is to inhibit the form of cytokine storm (Teng, Chen & Wang, 2021).

However, P. ternata diseases such as blight, tuber rot disease and viral disease occur frequently during its production. These diseases, caused by fungi, bacteria, viruses, can damage the leaves, stems, or tubers of P. ternata at all stages of growth. It has been reported that Choanephora cucurbitarum can cause flower blight disease in P. ternata (Wang, Mao & Tang, 2021), Pythium aphanidermatum can cause basal stem rot disease (Han et al., 2019), and Fusarium oxysporum (Sun, Hu & Liu, 2010) and Pectobacterium carotovorum subsp. Carotovorum (Shi, Li & Hu, 2015) can cause fungal and bacterial tuber rot diseases, respectively. These diseases seriously threaten the production of P. ternata. However, there have been few studies on leaf diseases in P. ternata. One study looked at Phytophthora parasitica Dast. causing leaf blight (Pei, Sun & Hu, 2010) and another at Alternaria alternate causing leaf spot (Wei, Chen & Wang, 2020). In recent years, spot blight disease in P. ternata occurred at a high frequency and diversity due to large-scale cultivation and continuous cropping. Spot blight disease seriously affects the photosynthesis and yield of P. ternata. Therefore, identifying the leaf spot pathogen is particularly important for the prevention and control of this disease.

In the summer of 2020, an outbreak of spot blight disease occurred in Anguo City, Hebei Province (N38°46′32″, E115°27′87″). Approximately 70% of plants there were infected by this disease, which greatly affected the yield and quality of P. ternata. This study aimed to identify the pathogens of spot blight disease on P. ternata based on their morphological and cultural characteristics, as well as molecular phylogenetic analysis.

Materials and Methods

Disease sample collection and pathogen isolation

Disease samples with typical spot blight symptoms were collected from three commercial fields in Anguo City (N38°46′32″, E115°27′87″), Hebei Province in August 2020. To isolate the pathogen, disease samples were sterilized with 75% alcohol for 4 min, then washed three times with sterilized distilled water. Samples at the junction of healthy and diseased areas were chopped into pieces (about 0.5 × 0.5 cm2), and then the pieces were plated on potato dextrose agar (PDA) medium containing cefotaxime sodium (100 µg/ml) and incubated at 27 °C in darkness. After the appearance of fungal colonies, hyphae tips were picked from the edges of the colonies with an inoculation needle for purification.

Pathogenicity test

One-month-old healthy P. ternata seedlings were grown in a controlled environment chamber under a 16 h light/8 h dark cycle at 25 °C ± 2 °C, relative humidity 85%. During pathogen inoculation, the healthy leaves and plants were wounded using syringe needles and infected with a 5 × 5 mm mycelial cake of pathogen, and sterile PDA disks were used as the control. The experiments were replicated three times, and a total of 30 seedlings were used. The incidence of spot blight was observed after three days. Fungi were recovered from the diseased leaves to complete Koch’s postulates.

Fungal identification

In this study, the isolated pathogens were identified using conventional morphological and microscopic characteristics. Pathogenic isolates were grown on PDA at 28 °C in darkness for 7–10 days to record colony morphology, color, and growth rate. The hyphae and spores were stained with cotton blue dye. And the size and features of conidia and chlamydospores were observed under a microscope (Olympus, Tokyo, Japan). The DNA of pathogenic fungi was extracted using the CTAB method (Freeman, Katan & Shabi, 1996). The rDNA internal transcribed spacer (ITS) region and 28s large subunit ribosomal RNA (LSU) were then amplified and sequenced using ITS1-ITS4 (Mitchell, Freedman & White, 1994) and LROR-LR5 primers (Moncalvo, Lutzoni & Rehner, 2000). PCR was performed in a 50 μL reaction system that contained 5 μL of 10× buffer, 1 μL of 10 mmol∙L−1 dNTP, 1 μL of 10 μmol∙L−1 forward primer, 1 μL of 10 μmol∙L−1 reverse primer, 1 µL of 5 U∙µL−1 DNA Polymerase, 1 μL of DNA and 40 μL ddH2O. The thermocycling program was as follows: 95 °C for 3 min, 34 cycles of 95 °C for 30 s, 55 °C for 30 s, 72 °C for 30 s, and a final extension of 72 °C for 5 min. The PCR products were sequenced and assembled by Tsingke Biological Technology Company (Wuhan, China). All sequences were deposited in GenBank under accession numbers MZ227385 and MZ227377 for ITS and LSU, respectively.

ITS and LSU sequences of other Stagonosporopsis spp. isolates were downloaded from the National Center for Biotechnology Information (NCBI) nucleotide database through BLAST. Alternaria brassicae was used as the outgroup. A phylogenetic tree was constructed using MEGA7 (Kumar, Stecher & Tamura, 2016) and the neighbor-joining (NJ) method (Saitou & Nei, 1987). The percentage of replicate trees in which the associated taxa clustered together in the bootstrap test (1,000 replicates) are shown next to the branches (Felsenstein, 1985). The evolutionary distances were computed using the Maximum Composite Likelihood method (Tamura, Nei & Kumar, 2004) and are in units of the number of base substitutions per site.

Statistical analysis

The statistical programme SPSS 18.0 (SPSS Inc., Chicago, IL, USA) was used to analyse the date. ALL results were confirmed by multiple biological repetitions.

Results

Disease incidence and symptoms

In this study, the initial disease symptoms were yellowish-brown spots on leaves that gradually expanded into irregular circular spots with brown centers and greenish-yellow halos surrounding the spots. The spot diameters ranged from 5–10 mm. These small spots connected into larger spots and eventually the entire leaf turned yellow and necrotic. Plants with severe disease also experienced death of all their aboveground parts (Figs. 1A and 1B).

Figure 1 Spot blight disease of P. ternata in field and Pathogenicity test in live plants.

(A and B) The phenotype of P. ternata spot blight disease in field. (C) Disease symptoms of P. ternata seedling at 7 days post inoculation with AG-3. (D) Disease symptoms of P. ternata seedling at 15 days post inoculation with AG-3.

Morphological characteristics of fungal isolates

A total of 15 fungal isolates were obtained from all diseased plant samples, 11 of them showed the same morphology and showed strong pathongenicity on the leaves of P. ternata in vitro. Among the other four fungi, two were Fusarium sp. and two were Penicillium sp. The pathogenicity of two different Fusarium isolates were very weak and the other two Penicillium isolates were also not pathogenic and were generally considered as miscellaneous fungi (Table S1). One strain from the 11 pathogenic isolates were selected and named as AG-3 for further study. The isolate AG-3 colonies grew on PDA for 7 days with a diameter of 60–75 mm at 28 °C. The colonies were regular, white to light gray in color, and had concentric rings seven days after culture. The color further deepened and the surface became gray black and the back became greenish-brown at 15 days (Figs. 2A and 2B). Conidia and chlamydospores formed after two weeks of growth and many small protuberances appeared on the surface of the colony. The conidia were hyaline and oval 4.6 to 8.7 × 1.2 to 2.4 µm (6.62 ± 1.24 × 1.87 ± 0.35, n = 30) in size, and most of them had diaphragms and contained small oil drops (Fig. 2C). Chlamydospores were unicellular, spherical to ellipsoid, 6.3 to 15 × 6 to 11 µm (11.52 ± 3.48 × 8.95 ± 1.64, n = 10) in size, and either single or 4–13 to a chain (Fig. 2D). Based on our morphological observations, the causal fungus was identified as Stagonosporopsis cucurbitacearum (Nuangmek, Aiduang & Suwannarach, 2018; Stewart, Turner & Brewer, 2015).

Figure 2 The morphology of AG-3 colony.

(A) The morphological characteristics of seven-day-old colony of AG-3 on PDA. (B) Twenty-day-old colony of AG-3 on PDA. (C) The morphological characters of conidia (100× magnification, dyeing treatment by cotton blue). (D) The features of chlamydospores chain of AG-3 (100× magnification, dyeing treatment by cotton blue).

Molecular identification

The ITS and LSU sequences of isolate AG-3 were uploaded to the GenBank database (accession numbers MZ227385 and MZ227377). BLAST results showed that all of the rDNA-ITS and LSU gene sequences of strain AG-3 showed 99% identity with the existing S. cucurbitacearum sequences in GenBank (JN618358.1, MK519412.1). Moreover, a phylogenetic tree of the ITS gene sequences of AG-3 constructed using the NJ method in MEGA7 software (Zhang, Qian & Zheng, 2019) revealed that AG-3 was closest to S. cucurbitacearum (Fig. 3). Based on morphological and molecular identification, the fungus was determined to be S. cucurbitacearum.

Figure 3 A maximum parsimony phylogeny tree for Stagonosporopsis sp.

Phylogenetic tree constructed with sequences of internal transcribed spacer ribosomal DNA (rDNA) region (ITS) of isolates AG-3 obtained in this study and other species retrieved from GenBank. The tree was constructed using the neighbor-joining method from the alignment of ITS sequences using MEGA software. Isolates from P. ternata are marked in bold.

Pathogenicity tests

For the pathogenicity test, three healthy, one-month-old P. ternata plants were infected with a 5 × 5 mm mycelial cake of AG-3. The other three control plants were treated with sterile PDA disks. The treatment group and the control group were placed in a culture room (25 ± 2 °C, relative humidity 85%). One week later, spot blight symptoms had developed on the pathogen-inoculated group, while no disease symptoms were observed in the control group (Fig. 1C). Two weeks later, the leaves of the infected plants had turned yellow and the plants died (Fig. 1D). Koch’s postulates were fulfilled by recovering pathogens from the inoculated plants that were reconfirmed as S. cucurbitacearum through molecular identification.

Discussion

S. cucurbitacearum was first reported in France and the United States but has now been isolated across the world (Chester, 1891). Previous studies found that the pathogen is an important disease for cucurbit crops and has been known to cause major yield and quality losses (Gao, Hao & Zang, 2020). S. cucurbitacearum can cause gummy stem blight disease on at least 12 genera and 23 species of Cucurbitaceae plants, including watermelon (Citrullus lanatus), cucumber (Cucumis sativus), and cantaloupe (Cucumis melo) (Keinath, 2011). S. cucurbitacearum can also cause serious damage to other economic plants such as Siraitia grosvenorii, water spinach, and tobacco (Jiang, Hu & Ye, 2015; Liu, Wei & Zhu, 2017; Wang, Zhou & Yu, 2018). This pathogen causes different disease symptoms on different tissues and organs. For example, if S. cucurbitacearum infects a stem, the diseased stem develops cankers with gummy exudate. In severe cases, the stem withers and the plant dies from stem canker or gummy stem blight. If S. cucurbitacearum infects the fruit, the diseased fruit shows black rot symptoms, and so it is called black rot. If S. cucurbitacearum infects the leaf, the diseased leaves show irregular spots with conspicuous yellow borders between the symptomatic and healthy tissues, and this is called foliar blight (Keinath, 2014; Keinath, 2000).

In recent years, S. cucurbitacearum has caused disease that affect the quality of plants used in TCM, such as Siraitia grosvenorii and Ningpo figwort (Zhang, Qian & Zheng, 2019). However, according to the evidence found so far, S. cucurbitacearum mostly infects the stems and fruits of plants more than the leaves. To the best of our knowledge, this is the first study on S. cucurbitacearum infecting P. ternata leaves in China. We observed S. cucurbitacearum causing the aboveground part of P. ternata to wilt, which seriously affected the plant’s yield and quality. This report will facilitate the diagnosis of P. ternata leaf spot, and corresponding measures must be adopted to manage this disease in a timely manner.

Conclusion

Using Koch’s postulates, we isolated the pathogen causing spot blight on P. ternata and identified it as S. cucurbitacearum. This is the first report on P. ternata spot blight caused by S. cucurbitacearum in China. Spot blight occurs rapidly, resulting in a huge yield loss. The occurrence of this disease should be closely monitored and preventative measures should be taken to avoid its spread. This study will provide valuable information for the prevention of Chinese P. ternata spot blight.

Supplemental Information

Supplemental Information 1 Morphology and pathogenicity of the 15 isolates.

Click here for additional data file.

Additional Information and Declarations

Competing Interests

Author Contributions

DNA Deposition

Data Availability

The authors declare that they have no competing interests.

Jia Zhou conceived and designed the experiments, performed the experiments, analyzed the data, prepared figures and/or tables, authored or reviewed drafts of the paper, and approved the final draft.

Jiawei Xu performed the experiments, prepared figures and/or tables, and approved the final draft.

Rong Xu analyzed the data, prepared figures and/or tables, and approved the final draft.

Qiaohuan Chen conceived and designed the experiments, prepared figures and/or tables, and approved the final draft.

Yunhan Wang performed the experiments, prepared figures and/or tables, and approved the final draft.

Bisheng Huang analyzed the data, authored or reviewed drafts of the paper, and approved the final draft.

Dahui Liu analyzed the data, authored or reviewed drafts of the paper, and approved the final draft.

Yuhuan Miao conceived and designed the experiments, authored or reviewed drafts of the paper, and approved the final draft.

The following information was supplied regarding the deposition of DNA sequences:

The sequences are available at Genbank: MZ227385 and MZ227377.

The following information was supplied regarding data availability:

The raw measurements are available in the Supplemental File.

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
