# Peer review of "Identifying and characterizing Stagonosporopsis cucurbitacearum causing spot blight on Pinellia ternata in China"

_PeerJ, doi:10.7717/peerj.13278_

## Round 0.1 · original submission · Major Revisions

Dear authors:

Two experts on the subject have assessed your manuscript. Although one reviewer recommended rejection, after careful assessment of the comments and concerns raised by the reviewer and the comments done by the second reviewer, I think all the concerns can be attended and presented in a new version of the manuscript.

I kindly request that you carefully check all the comments and modify your manuscript accordingly. I think the information presented here is of value and worth amending.

I hope we can receive your revised version of this manuscript.

With best regards

Reviewer 1 ·

Basic reporting

-In general, clear and unambiguous, professional English is used throughout
-Literature references are adequate, but some of them are incomplete:
Guideline 184. Reference is not found in the literature
guidelines 189,191,194,209 and 234. References are incomplete. Missing authors
Guidelines 197, 200, and 203. References are not ordered by year.
-Professional article structure is correct. Description of the feet of the figure is poor. The microscopy images do not describe the staining techniques used in Figure 2.

Experimental design

Methods
Guidelines 85 to 87.
-The description of the methods is not clear.
-There is no complete information on the staining techniques used nor the amplification conditions of the PCR technique volumes of each reagent used are given, but not their concentrations.
Guideline 85.
-Primers are not well described. In the cited reference, they do not use the initiator LR5; they use LR7.
Vaghefi, N. and Pethybridge, S. J. and Ford, R. and Nicolas, M. E. and Crous, P. W. and Taylor, P. W. J. (2012) Stagonosporopsis spp. associated with ray blight disease of Asteraceae. Australasian Plant Pathology, 41 (6). pp. 675-686. ISSN 0815-3191
Methods are not sufficiently detailed for replicability
Results
Guideline 111.
-Fifteen fungi were isolated, 11 strains were the same organism (data not shown), and the other four were not results?
Guidelines 111 & 112
-describe one dominant strain? What characteristics of the AG-3 strain define it as dominant?
-In Materials and Methods, in the part of molecular identification of the fungus, in guideline 95, they say that they used Fusarium oxysporum as an external group in the phylogenetic tree, while in the results (Figure 3) in the phylogenetic tree, the fungus of the external group used is Alternaria brassicae.
Guideline 118.
-Have tiny oil droplets; how does it know? What staining technique did it use?
-There is no statistical analysis or results
-it is an interesting article because it describes for the first time the presence of the pathogen in the leaves of the plant, but the presentation of the results is more like a disease note.

Validity of the findings

-It is important to determine with precision the pathogenic fungus that is affecting the plant, in order to propose fungus control strategies, but it is not the only fungus they detect on the leaves of the plant.
-The 15 fungi that were isolated must have the same phenotype, and only one strain, AG-3, was analyzed, it is not clear why only strain AG-3 to used. Not discussed what happened with the other four strains.

Additional comments

the manuscript has errors in the description of the methods used and in the references cited that should be corrected.

Reviewer 2 ·

Basic reporting

The manuscript is clearly written with a few editorial errors e.g. spacing issues. Specifically, some comments I have include:
Line 32 Add space before '(Thunb.)'
Line 38 Remove space between Chinese and Lung
Line 39 …to treat the symptoms of COVID-19
Line 72: Since this is the first mention of the isolate name (AG-3), it should be mentioned at this point that the pathogen was named AG-3.
Line 144: add space between S. and cucurbitacearum

Experimental design

Avoid repeating materials and methods in the Results section e.g.
Line 103-104: This statement should not be included on the results section

Validity of the findings

Line 111: Of the 15 fungal isolates, what were the other 4 that were different? This information should be included.

Brewer et al., 2015 provides PCR-based primer sequences useful for distinguishing 3 Stagonosporopsis species. Test these primers on AG-3 and the other 4 isolates to further confirm the S. cucurbitacearum bands and include that information on the manuscript.

---

## Round 0.2 · accepted · Accept

Dear authors,

After a second round of review and revising the manuscript myself, the present manuscript is suitable for publication. I personally thank you for addressing the concerns raised by the reviewers.

Thank you for choosing PeerJ.

With best regards,
Bernardo

Reviewer 2 ·

Basic reporting

Following the changes made by the authors, the manuscript is okay for publication.

Experimental design

No comment

Validity of the findings

No comment